# Use of Gamma Radiation for the Genetic Improvement of Underutilized Plant Varieties

**DOI:** 10.3390/plants11091161

**Published:** 2022-04-26

**Authors:** María de la Luz Riviello-Flores, Jorge Cadena-Iñiguez, Lucero del Mar Ruiz-Posadas, Ma. de Lourdes Arévalo-Galarza, Israel Castillo-Juárez, Marcos Soto Hernández, Carlos Roman Castillo-Martínez

**Affiliations:** 1Colegio de Postgraduados, Campus Montecillo, Km. 36.5, Carretera México-Texcoco, Montecillo, Texcoco 56230, Mexico; marluriviello@gmail.com (M.d.l.L.R.-F.); lucpo@colpos.mx (L.d.M.R.-P.); larevalo@colpos.mx (M.d.L.A.-G.); israel.castillo@colpos.mx (I.C.-J.); msoto@colpos.mx (M.S.H.); 2Colegio de Postgraduados, Campus San Luis Potosí, Salinas de Hidalgo, San Luis Potosí 78622, Mexico; 3CENID-COMEF, Instituto Nacional de Investigación Forestales, Agrícolas y Pecuarias (INIFAP), Progreso Núm. 5, Barrio de Santa Catarina, Alcaldía Coyoacán 04010, Mexico; castillo.carlos@inifap.gob.mx

**Keywords:** in vitro crop, phytochemicals, gamma radiation, ionizing radiation, mutants

## Abstract

Agricultural biodiversity includes many species that have biological variants (natives, ecotypes, races, morphotypes). Their use is restricted to local areas because they do not fulfill the commercial requirements; however, it is well documented that these species are a source of metabolites, proteins, enzymes, and genes. Rescuing and harnessing them through traditional genetic breeding is time-consuming and expensive. Inducing mutagenesis may be a short-time option for its genetic improvement. A review of outstanding research was carried out, in order to become familiar with gene breeding using gamma radiation and its relevance to obtain outstanding agronomic characteristics for underutilized species. An approach was made to the global panorama of the application of gamma radiation in different conventional crop species and in vitro cultivated species, in order to obtain secondary metabolites, as well as molecular tools used for mutation screening. The varied effects of gamma radiation are essentially the result of the individual responses and phenotypic plasticity of each organism. However, even implicit chance can be reduced with specific genetic breeding, environmental adaptation, or conservation objectives.

## 1. Introduction

Beneficial mutations are changes to the genotypic structure that increase the variability of the species and favor their adaptation to various selection pressures [1]. These can be induced by physical mutagenic agents such as ionizing radiation (X-rays and gamma rays), non-ionizing radiation (ultraviolet), and corpuscular radiation (protons, neutrons, alpha, and beta particles) [1,2]. Ionizing radiation (IR) induces the change from neutral molecules or atoms to their ionized forms; this change requires ionization energy, which is the minimum amount of energy that separates the electron from a free atom in its lower energy state [1,2], through two effects: Compton and photoelectric [3].

IR can directly induce physical, biological, and chemical changes in the cells altering the chemical nature of the molecules [1,2,4]. It can induce specific changes in the genome [5] and indirectly induce an alteration of free radicals generated mainly by the ionization of water molecules [1,2,4] (Figure 1).

Ionizing energy has been used for approximately 60 years to develop new varieties with economic value, improved resistance to pests and disease, higher agricultural yield, better quality, and high nutritional value, among other advantages. Population growth, food shortages, the economic situation of some regions of the world, the excessive use of non-renewable energy resources (fossil fuels), and climate change have led to the search for new uses and the genetic improvement of underutilized vegetables [10]. So-called underutilized plant species varieties (neglected, orphan or minor) are an option to substitute crops with broad commercial demand [9]. Although they are considered popular crops for local consumption or self-consumption (not commercially competitive), they represent a potential option for diet diversification, owing to their wide possibility of improvement, processing, and adaptation to the environment [9]. This review focuses on the research about plant breeding mediated by GR in recent years and its relevance to obtain good agronomic characteristics for agriculture. In addition, future perspectives for its application in underutilized genetic varieties—as an alternative for the use of little valued genetic resources—is highlighted.

The information search was carried out in Google Academic, Scopus, and SciELO. The following search words were used: “ionizing radiation in plants”, “the effect of gamma irradiation on secondary metabolites”, “induced mutation by gamma irradiation”, “effect of ionizing radiation on DNA-plant”, “plant mutation breeding”, “in vitro mutagenesis”, and “secondary metabolites”. The mutant varieties database of the International Atomic Energy Agency (IAEA) was also consulted, using the search criteria “physical agent” and “gamma radiation”.

## 2. Radiosensitivity

Efficient induction of mutagenesis by gamma radiation (GR) requires determination of the optimal radiation dose; that is, the dose that reduces 50% of the population (median lethal dose, LD_50_). It also requires variables such as: survival, mass, or number of germinated specimens, among others; or, the radiation dose that reduces growth in 50% of the population (median growth reduction, GR_50_). Both doses depend on the plant tissue (seed, meristem, callus, etc.), stage of development and moisture content, among other parameters [11,12]. High radiation doses can induce radioinhibition by affecting growth promoters and eventually tissue destruction [13]. It can also cause loss of regenerative capacity and malformation of plant tissues as well as tissue destruction [14]. Radiosensitivity assays allow determining the appropriate radiation dose to induce the highest mutation rate with the most negligible effects on the gene complex [1,15]. Radiation stimulation can be obtained with low radiation doses that favor the induction of metabolites and biochemical changes involved in plant regeneration [13,16], thus generating chimeric plants [13]. Some studies have reported that radiation stimulation by GR is demonstrated in plant varieties. Kaolack and Crimson irradiated sweet varieties of *Citrullus lanatus* (watermelon) inducing an LD_50_ of 225.40 and 221.56 Gy, respectively [17]. While in *Coffea arabica* L. var. *typica* with an LD_50_ of 100 Gy, morphological changes were induced in the leaf (color, number, length, and width), plant height, and distance from the cotyledon to the first node [18]. An LD_50_ of 2486 Gy was recommended for the *Eragrostis superba* (Wilman lovegrass) plant [12], while 628, 712, 698, and 411 Gy were recommended for the seeds of *Eragrostis curvula* (weeping grass), *Pennisetum ciliare* (Buffel), *Bouteloua curtipendula* (Banderita), and *Bouteloua gracilis* (Navajita), respectively [19]. In corns and seedlings of *Agave tequilana* var. *blue*, a LD_50_ between 20 and 25 Gy induced the number of shoots and increased the size of the seedlings, while 16 Gy increased the callus area [20]. For *Solanum tuberosum* var. Désirée (potato), an LD_50_ of 10 Gy increased the regeneration of vegetable callus by 71% [21]. Meanwhile, GR-induced DNA breaks the form of harmful genomic and chromosomal abnormalities [22]. The erroneous nature of DNA repair can lead to aberrations that can be latently transmitted to the progeny [23]. Some examples are reported in the axillary buds of *Physalis peruviana* (Golden Berry/Uchuva): at 100 Gy cytological changes were induced and the number of lagging chromosomes increased, while at 200 Gy, the formation of anaphase, telophase, and isolated chromosome bridges increased [24]. The alterations during anaphase revealed that GR influences the function of the mitotic spindle; the bridges are caused by the formation of dicentric chromosomes that originate from the chromosome exchange after DNA breakage [25]. Another example can be found in *Chrysanthemum morifolium* “Donglinruixue” (chrysanthemum): a 35-Gy radiation increased the number of cells with chromosomal aberrations (adhesion, univalent, unipolar, lagging chromosomes) and the formation of micronuclei [26]. In *Saccharum* spp. hybrid var. “*SP 70-1284*” (sugarcane) doses higher than 30 Gy compromise callus growth and regeneration [27]. In general, ionizing radiation-induced DNA repair uses mechanisms similar to those that regulate the integrity of foreign sequences in plant genomes [5,23]. It can occur by a double-strand break (DSB) repair route:the first recovers information through homologous recombination (HR), and the second uses a non-homologous end-joining (NHEJ) system that often results in the deletion of a DNA fragment and therefore a mutation [9]. The adaptive and developmental process of plants provides them with greater resistance to the production of DSB by ionizing radiation than other eukaryotic organisms, triggering much more effective and rapid DNA repair and defense reactions.

## 3. Gamma Radiation and In Vitro Culture

In vitro mutagenesis in a plant organ, tissue, and cell cultures provides the opportunity for the rapid and massive spread of mutant plants [1]. The induction of mutations through GR has been reported in buds and seeds collected in the field, as well as in tissue cultures, protoplasts, and callus [28]. With this technique, large populations of plants can be managed under controlled conditions, in reduced spaces, and at any time of the year [9]. On the one hand, in vitro biotechnological techniques guarantee quality and safety, which combined with mutagenesis techniques, allow the selection of outstanding agronomic characteristics in plants of economic and productive importance [29]. On the other hand, although the aim is to obtain improved characteristics with GR, negative responses on growth have also been reported, as is the case of *Gerbera jamesonii* calluses in which an 11% reduction of fresh weight was recorded after subjecting them to 20 Gy from GR [30]. Therefore, using wide ranges of GR to cover the radioinduction and radioinhibition intervals must be taken into consideration. Also, the basic physiological and taxonomic aspects and the oxygen and water content of the plant material to be irradiated must be determined. In this regard, the effect of low GR doses on the growth and accumulation of 20-hydroxyecdysone in *Sesuvium portulacastrum* tissue culture has been reported. When the dose of 20 Gy doubled the concentration of the metabolite (0.139 mg/dry weight), the GR was proved to induce metabolic changes and obtain mutants with a higher in vitro production of secondary metabolites [31]. Another research reported the effect of 15–45 Gy doses combined with monochromatic lights on the growth of in vitro shoots of the *Dendrobium sonia* orchid. GR decreased shoot length, fresh weight, and leaf area, but its combination with yellow light increased shoot survival and length, fresh weight, and chlorophyll content [32]. Similarly, using different GR doses (10, 30, and 50 Gy) on in vitro cultured explants of *Eriobotrya japonica* L. (Loquat) made it possible to accelerate the reproduction programs in which the combination of apical explants grown in MS medium with 10 Gy of GR favored a greater growth of callus diameter and height, the number of leaf shoots, and seedling height [33]. Likewise, good results have been reported in species of difficult reproduction. For example, the vegetative cycle of *Musa paradisiaca*—which reproduces vegetatively—requires a long generational time; this makes genetic variation difficult. The effect of radiation on the diversity of the Pisang Ambon crop was determined, when 10 and 20 Gy were found to increase the length and width of the leaves, as well as the height and diameter of the stem. These results are important, because they support the massive propagation of banana plants of better quality and with greater resistance to biotic and abiotic stress [34]. Additionally, positive effects of GR have been reported on different plant structures of various varieties of *Citrus* spp. However, since all the material was susceptible to GR-induced mutagenesis, the seeds showed the highest resistance (LD_50_ at 127 Gy in “Alemow”, and 156 Gy in sour orange), followed by the buds (LD_50_ around 50 Gy for all cultivars) and the nodal segments (LD_50_ of 25 Gy for both lemon cultivars) [35]. Therefore, the GR applied to in vitro cultures allows the development of multiple responses in plant materials, either to promote variability in a short time in species where it is difficult to obtain it traditionally or to enhance the metabolic expression.

## 4. Gamma Radiation as a Tool for Plant Breeding

The induction of mutations by physical agents allows to obtain genetic variations in crops of agronomic importance that are not found in nature [35]. The alteration caused by the formation of free radicals promotes structural and metabolic changes in the plant [10]. For example, studies about the effect of GR on chloroplasts show that they were mainly sensitive to high radiation and that mutations occurred after 50 Gy [36]. In addition, high doses of GR affect protein synthesis, hormonal balance, enzymatic activity, and gas and water exchange [36]. Currently, there is great interest in generating new variations to develop more nutritious, resilient, and productive crops [37,38].

Although the generation of new varieties by GR is random, they can be identified and selected in a short time, accelerating the establishment and release process; compared to the process used to obtain Genetically Modified Organisms (GMOs) this is a more efficient method [1]. According to the mutant varieties database of the International Atomic Energy Agency (IAEA), from 1960 to 2020, 1703 plants of agronomic importance with GR-induced mutations were recorded. Specimens such as the Rosa mutant (Pink Hat)—generated by radiating terminal buds—were approved in 1960; their advantages include a two-tone flower color and resistance to mold [39]. Recently, *Oryza sativa* L., *Solanum lycopersicum* L., and *Bougainvillea* spp. mutants were registered [39]. In the case of *O. sativa* (rice), the organoleptic characteristics, yield, and early maturation, as well as their resistance to *Nilaparvata lugens* infection increased in the mutants generated with GR [39]. When the seeds of *S. lycopersicum* (tomato) were subject to radiation, mutants with higher yields and fruit quality (shape, size, and color of the leaves) were obtained; they also have a greater tolerance to heat stress. As a consequence of its success, one of its varieties has been transferred to the industrial sector, where it is used to produce different food products [39]. Also, it was reported that doses of 10 and 20 Gy of GR increases the stomatal parameters (density, length, and width of the stomata) of three varieties of sugarcane (NIA-0819, NIA-98, and BL4), which has an impact on gas exchange and photosynthetic activity [40]. In *Triticum aestivum* L., radiation on the seeds of two genotypes (Roshan and T-65-58-8) decreased the number of shoots, the length of the root, and the dry weight of the seedlings, without affecting germination. However, when the seedlings were irradiated with 100 Gy, proline and chlorophyll A content increased by 85%. Therefore, the expressed phenotype depends on the radiation dose, which may enhance resistance to radiation, drought or saline stress [36].

With GR, mutants resistant to heavy metals, salt stress, and pests can also be obtained. *Hordeum vulgare* L. (mountain barley) seedlings irradiated with 50 Gy and subjected to lead and cadmium stress presented less hydrogen peroxide and malondialdehyde. In addition, a high concentration of proline and antioxidant enzymatic activity suggested the expression of heavy metal transporters [41]. The effect of GR on the induction of resistance to saline stress was reported in *S. tuberosum* L. (potato) and *O. sativa* L. (rice) cultures, in which several mutants resistant to high concentrations of salt were generated [42,43]. On the one hand, three lines of *Cicer arietinum* L. (chickpea) were also induced with GR (LD_50_ of 150 Gy) and became resistant to *Ascochyta rabiei* (chickpea blight) [44]. On the other hand, GR has also been used in difficult-to-handle cultivars with long harvest times: for example, Mentik Susu, a variety of rice that is very difficult to harvest, as a consequence of its great height. Using GR (200 Gy), smaller, more productive mutants with a better seed yield were obtained; harvest time was also shorter [45]. Finally, mutants with higher commercial quality characteristics can be obtained with GR—for example, *Pamindo agrihorti* (grapefruit), derived from *Citrus maxima*. The stability of the characters and greater vigor were also demonstrated.

## 5. Effect of Gamma Radiation on the Concentration of Secondary Metabolites

Secondary metabolites are organic compounds with low molecular weight; they are biosynthesized from or share substrates of origin with the primary metabolism. They include phenolic compounds, terpenoids, and compounds that have nitrogen in their structure [46]. They are of utmost importance because they help the plant to interact with biotic and abiotic factors in the hostile environment in which they develop, creating defense and protection mechanisms [46]. In some cases, the secondary metabolites are highly specific to the species, providing them with important qualities for their application as therapeutic or medicinal agents [46]. One area of interest is the profitable production of secondary metabolites through biotechnological mechanisms and mutagenesis. Under this approach, the GR technologies that have been applied increase the concentration of phytochemicals of interest in both sexually or vegetative propagated (in vitro) plants of commercial or medicinal value. GR interacts with the biosynthetic mechanism of the plant forming free radicals (ROS) caused by the radiolysis of water. This triggers the oxidative stress response, producing various defense enzymes and antioxidants [47], which is proportional to the concentrations of secondary metabolites. It is important to highlight that a low GR dose generates accelerated cell proliferation, germination rate, cell growth, enzymatic activity, resistance to stress, and increased crop yield and contraction of secondary metabolites [31,47].

When the calluses of *Rosmarinus Officinalis* L. (Rosemary) are irradiated, the activity of phenylalanine ammonium lyase (PAL) increases and, at the same time, the concentration of total phenols also increases. This enzyme is key in the pathway of the phenylpropanoids responsible for the synthesis of phenolic acids [48].

Moghaddam et al. [49] demonstrated that, after eight weeks, the seedlings of the accession CA23 of *Centella asiatica* irradiated with 20 and 30 Gy doses contained the highest concentrations of total flavonoids (16.827 ± 0.02; and 16.837 ± 0.008 mg g^−1^ dry weight, respectively): 54.7% and 46.8% more than control.

Flavonoid biosynthesis is stimulated by enhancing the phenylalanine content, the phenylalanine ammonia-lyase (PAL) activity, and the chalcone synthase enzyme (CHS) activity. Specifically, regarding the response to gamma radiation, PAL activity affects flavonoid synthesis in the phenylpropanoid pathway, where this enzyme acts as a catalyst for the conversion of phenylalanine to cinnamic acid. However, the enzyme chalcone synthase (CHS), key in the biosynthesis of flavonoids or isoflavonoids, also catalyzes the formation of chalcones from malonyl-coA and coumaroyl-coA. Chalcones are a major intermediate in the flavonoid pathway that gives rise to several classes of flavonoids. There is a correlation between the upregulation of CHS genes and increased flavonoid content in response to gamma radiation [49].

In other researches, seeds of *Onobrychis viciifolia* Scop were irradiated. The leaf extract of Syn. *Onobrychis sativa* L. (Sainfoin) irradiated with 90 Gy had a greater phenolic content than non-irradiated extract. Likewise, the alkaloid Berberine increased from 0.000152% to 0.000203% [50].

Sheikhi et al. [51] studied the effect of gamma radiation on calluses of *Ferula gummosa* Boiss. and found that the phenolic content increased by 36.5% and 38.9%, when they were irradiated with 20 and 25 Gy doses.

It should be mentioned that the effect of gamma radiation on the concentration of terpenoids, alkaloids and other metabolites of medicinal importance has also been studied. Muhallilin et al. [52] researched the diversity of morphological characteristics and chemical content of the seedlings of *Celosia cristata* L., an ornamental plant valued for its application in traditional medicine. They selected a clone (labeled C1U3 2.3.1) irradiated with 25 Gy dose, which showed a remarkable triterpenic compounds content that was not present in the controls.

Magdy et al. [53] identified that the M2 offspring of a rhizome of *Zingiber officinale* Rosc (parental ginger) irradiated with 20 Gy had 73.76% more 6-gingerol content than the control: 38.4 ± 0.01 mgg^−1^ of methanol extract compared to 22.1 ± 0.03 mg g^−1^ of methanol extract in the non-irradiated control samples.

Kapare et al. [31] studied the impact of low doses of gamma radiation (range: 5–40 Gy) on the growth and accumulation of 20-hydroxyecdysone in *Sesuvium portulacastrum* shoots. They verified that ex vitro plants obtained from shoots exposed to 20 Gy had 66% more ecdysteroid 20-hydroxyecdysone content (0.321 mg dry weight of the plant^−1^) than the control.

Azeez et al. [54] studied the effect of gamma radiation on the yield of pharmacologically-profitable secondary metabolites in callus cultures obtained from the leaf, stem, and root of *Hypericum triquetrifolium* Turra, irradiated with doses of 10, 20, 40, and 50 Gy. They specified that the best irradiation doses that stimulated epicatechin concretion were 10 and 20 Gy in leaf and stem callus (126.39 and 148.80 mg 100 g^−1^) regarding the control (98.81 and 101.72 respectively). Likewise, they registered that the foliar callus irradiated with doses of 10 Gy had higher naphthodianthrones hypericin and pseudohypericin content than the control samples (hypericin 10 Gy: 0.294 mg 100 g^−1^; control: 0.251 mg 100 g^−1^; pseudohypericin 10 Gy: 4.01 mg 100 g^−1^; control: 3.57 mg 100 g^−1^).

The concentration of alkaloids such as trigonelline and nicotinic acid (two secondary metabolites of medicinal importance) was significantly affected by the gamma radiation incidence on *Trigonella foenum-graecum* L. (fenugreek) seeds. In this research, the highest result was obtained with the 100 Gy dose, with 7% and 9% increases of these secondary metabolites compared to the control [55].

Mariadoss et al. [56] applied gamma radiation to obtain high-performance cell cultures of *Rubia cordifolia*. Callus irradiated with 8 Gy accumulated a maximum alizarin and purpurin levels that were 6 and 11 times higher than non-irradiated callus cultures. From the mutants, a suspension cell bioreactor protocol was established, obtaining 63.58% more anthraquinones.

Overall, the application of GR technology to increase secondary metabolites such as those mentioned and others—capsaicinoids [57], steviosides [58], saponins, ginsenosides [59], camptothecin [60]—offers an impressive study window about various plant species with medicinal applications; at the same time, it provides an opportunity to move towards a more sustainable agriculture. Therefore, further research about the potential for the application of technologies —such as ionizing radiation and biotechnological protocols—are necessary. When they work together, the success in the objectives set increases.

## 6. Molecular Analysis for the Identification and Screening of Mutants

The genetic breeding process through physical agents such as GR is strengthened when a molecular identification of specific agronomic traits is carried out; this allows the screening of DNA polymorphisms between mutants or point mutations. This screening identifies whether the mutations are the results of single base substitutions, deletions, insertions, inversions, duplications, reciprocal translocations, among others. Meanwhile specific researches obtained useful plant material for the application of functional genetics. Whole genome resequencing, molecular markers (such as ISSR, RAPD-PCR), or more complex methodologies (such as QTLs, TILLING or transcriptomics supported with bioinformatic analysis) can be used to achieve these objectives.

Magdy et al. [53] recorded some examples of these molecular methodologies, which they used to obtain variability in curl-mas of *Z. officinale* Roscoe (ginger), confirming it through RAPD-PCR analyzes; they used five primers and obtained 58 bands; 15 monomorphic and 43 polymorphic (74.14% polymorphism). The polymorphism levels varied with each primer, confirming the genetic variation in the plants irradiated with gamma rays compared to the control.

ISSR (Inter Simple Sequence Re-peats) markers were used to identify DNA polymorphism among gamma-radiation mutants of *Sophora davidii* (Franch.) Kom. ex Pavol (medicinal plant and food scrub of ecological value). This analysis allowed the generation of 183-point fragments with 51.37% polymorphism. The genetic similarity based on the ISSR data ranged from 0.6885 to 1.000 (Jaccard coefficients of dissimilarity), with an average genetic similarity of 0.7884, which indicated the level of genetic variation between mutants. Therefore, gamma ray treatment proved to be an effective way to induce mutations in *S. davidii* and that mutants were successfully detected by ISSR analysis [61].

Li et al. [62] characterized GR and ion beam-induced mutations in *Oryza sativa* L. mutant lines of the M5 generation by whole genome resequencing and bioinformatic analysis. Fifty-seven single-base substitutions (SBS), 17.7 deletions, and 5.9 insertions were detected in each irradiated mutant. An analysis of structural variation (SV) was performed and an average 0.6 SV (spanning large deletions or insertions, inversions, duplications, and reciprocal translocations) were detected in each mutant.

2-Acetyl-1-pyroline (2AP) is a volatile compound responsible of the aroma in rice and it is biosynthesized when the BADH2 gene loses its function as a suppressor gene. Aromatic rice cultivars naturally incur the BADH2 gene mutation at 8 bp. Some homozygous mutant rice lines were obtained by gamma radiation with a 100 Gy dose and various aroma-related primers of rice were used to identify the point mutation. PCR was performed and 254-bp and 355-bp DNA fragments were sequenced to identify the genetic mutation. The nucleotide sequence data of these DNA fragments showed that point mutations (deletions and substitutions of purine for pyrimidine or vice versa) occurred in the BADH2 gene in exon 7; these are called second mutations and were caused by gamma rays [63].

Tan et al. [64] characterized mutations in coding regions of a *Hordeum vulgare* (barley) dwarf mutant induced by gamma radiation, using a transcriptome sequencing strategy. They found 1193 genetic mutations in gene transcription regions: 97% of these were concentrated in the 5H and 7H chromosome regions. They also found that the mutations were not uniformly distributed throughout the genome, but that they were located in several concentrated regions. This is a clear example that provides a deeper understanding of the mechanisms of gamma radiation mutation and its application in the analysis of genetic function.

Regarding the monitoring of induced mutations in cultivars of agronomic and economic importance, outstanding molecular methodologies have been developed, including QTL (identification of quantitative trait loci). In other cases, the outstanding application of gamma radiation to induce mutations enables important research lines for the application of functional genomics, such as the creation of TILLING populations.

TILLING is a reverse genetics methodology in functional genomics research that helps the functional identification of mutations in specific genes [65]. It has been used together with gamma radiation to obtain cultivars with outstanding agronomic characteristics. To achieve this procedure, approximately 15,000 m^3^ of cultivars were developed by applying gamma radiation to rice seeds (*O. sativa* subsp. Japonica cv. Donganbyeo). The salient characteristics in the TILLING population were analyzed, using AFLP molecular markers and evaluating genetic diversity. Subsequently, 28 polymorphic loci of the TILLING lines were cloned. Overall, this study proposed the TILLING rice population as a valuable genetic source that can be used in functional genomic studies about the species [66].

Continuing with the use of more sophisticated methodologies for the screening of gamma radiation-induced mutations, Hwang et al. [67] researched the QTL to determine the flowering time of a rice mutant obtained by gamma radiation, whose importance is associated with crop yield and quality. To achieve their objective, they developed a linkage map of 36 InDel markers and six SNP markers with F2 plants derived from the “WT 9 EMT1” cross. They detected a main QTL region in chromosome 6 and a candidate gene to control the early heading date in EMT1 by genetic linkage analysis, sequence variation and expression study. The results obtained suggested that the genes related to the temperature-sensitive flowering pathways could promote the regulation of flowering, such as the EMT1 gene, which provides the clearest explanation for the flowering mechanisms of rice under LD conditions and for the development of new early flowering rice cultivars.

Recently, technologies such as transcriptomics have been used to provide a specific response to the effect of gamma radiation on the genome. Kang et al. [68], researched the gene expression changes of *Vigna unguiculata* (L.) Walp. (cowpeas) plants subjecting them to different doses of ionizing radiation: gamma radiation and proton beam. They identified differentially expressed genes (DEG) in the entire genome of the irradiated plant through the classification of the genes; subsequently, they were able to make a general description of the metabolic pathways that were involved in the stimulation of change in the plant. The response to irradiations doses was diverse: three genes were identified and expressed differently with each irradiation dose (32, 75, and 69) than control. In contrast with control, 168, 434, and 387 DEG were identified for each proton beam irradiation dose. The number of genes related to defense, photosynthesis, reactive oxygen species (ROS), plant hormones, and transcription factors (TF) that were up- or down-regulated was higher in proton beam treatment than in gamma ray treatment.

Kim et al. [69] observed a 208-lines gene expression pattern in a group of mutant diversity (MDP) of M12-generation *Glycine max* L. (soybean), obtained by gamma radiation; they selected and studied the metabolic properties observing the isoflavones and fatty acids content of the seeds. Six lines with altered isoflavone content and six lines with altered oleic acid content were selected and compared with wild types in order to measure gene expression. The isoflavone biosynthetic genes were different in each stage and expression patterns and, in the mutants that presented a higher concentration of isoflavones, the MaT7 gene showed a higher expression level. Fatty acid biosynthetic genes were classified into two groups that reflect the development stages of the seeds. Consequently, the bases were established for a future functional analysis of the regulatory genes that are involved in the biosynthetic pathway of isoflavones and fatty acids in soybeans.

## 7. Prospects for the Application of Gamma Radiation in Underutilized Genetic Varieties

Plant genetic resources for food and agriculture (PGRFA) are also considered as part of agricultural biodiversity. These are defined as plant material (including reproductive and vegetative propagation) with actual or potential value for food and agriculture [70]. Currently, thirty main crops are used commercially to supply the global food demand, but only wheat, rice, corn, and potatoes (13.3%) are classified as food security [71,72]. Meanwhile, approximately 23% of the 30,000 edible plant species are collected and used as food. However, although they can diversify the diet with new nutritional sources, there have been few studies and evaluations in this regard. Since they lack commercial interest, some species have been cataloged as underutilized (forgotten, obsolete, or minor) and their nutritional and cultural properties, as well as their role as a source of genetic diversity, have not been valued. It is important to note that many of these plants are used in local agriculture and they have been the basis of community food for centuries. However, although commercially important crops have replaced many of them, they have now been incorporated into genetic improvement studies [73].

Padulosi et al. [74] mention that underutilized species depend on the geographical area, as well as the social and economic impact. In certain places, the species is well known and widely used, while in others, it is classified as underutilized. An example is *Cicer arietinum* (chickpea), which is in Italy considered an underutilized species, but which in Syria is a staple food and the basis of their diet.

Other species are the vegetables *Eruca sativa*, *Diplotaxis tenuifolia*, and *D. muralis* (arugula), which have significant economic value in Europe; meanwhile, in Egypt, they provide nutrients to the poorest population and are therefore unexpensive and locally used [74].

Cateano et al. [73] mention some crops with different marginality categories in countries such as Ecuador, Brazil, Bolivia, Spain, Peru, Colombia, and the Andean Region. Furthermore, they identify many orphan, obsolete, promising, and underutilized crops in Colombia.

Bravo, Arteaga and Herrera [7] recorded 91 species used for food and medicinal use in northern Venezuela. Forty-six of these species are known as alternative or underutilized plants, they are used for self-consumption, their commercialization is very limited, many of them exist in the wild, and they have the potential to strengthen the local diet. This group includes *Annona cherimola* (custard apple), *Annona squamosa* (Anón), *Spondias mombin* (Jobo), *Spondias purpurea* (stone plum), *Chrysophyllum cainito* (caimillo), *Coleus forskohlii* (oregano), *Curcuma longa* (turmeric), *Portulaca oleracea* (purslane), *Eryngium foetidum* (wild coriander).

In Mexico, some research centers have undertaken to conserve and make use of the PGRFA, rescuing several crops that are a priority for food sovereignty whose basis is the underutilized biological variants (Table 1). The scheme integrates the rescue, conservation, characterization, and development of capacities following the Second World Plan of Action for Plant Genetic Resources for Food and Agriculture of the FAO [75,76]. Forty-four networks were established per crop and a Thematic Network of Conservation Centers, carrying out actions in four strategic areas and eighteen lines of action.

Genetic resources are a reservoir for conducting bioprospective studies (Table 1) and identifying new sources of nutrients and compounds that provide health benefits. They also offer a potential solution to the growing demand of the food industry and favor its protection as a material that generates value for local communities.

Although they have been undervalued, the group of plants known as quelites is currently classified within the Priority Biocultural Regions of Mexico and it is linked to the Great Geoeconomic Regions of the country [75]. Approximately 127 native herbaceous species that grow in the *milpas* fall within the denomination of quelite, out of which 12 species are the most representative, including: *Portulaca oleracea* (purslane), *Amaranthus* spp. (amaranth), *Dhysphania ambrosioides* (epazote), *Porophylum ruderale* subsp. *macrocephalum* (papalo), *Chenopodium berlandieri* (quelite), *Solanum americanum* (blackberry), *S. nigrescens* (divine nightshade), *Anoda cristata* (alache), *Jaltomata procumbens* (jaltomate), *Lepidium virginicum* (virgina pepperweed), and *Phytolacca icosandra* (amolquelite) [76].

On the one hand, quelites are semi-domesticated plants of local and traditional consumption. Its production has been remained under the protection of small producers; additionally, several quelites have agronomic qualities such as resistance to drought and low maintenance [76]. They also have great commercial potential, including. *ruderale* subsp. *macrocephalum*, *P. oleracea*, *Amaranthus* spp., *Suaeda edulis* (romerito), and *Chenopodium berlandieri* subsp. *nuttaliiae* (huauzontle); they are produced on a large scale, with national and export demand [76]. Therefore, quelites are a clear example that underutilized crops can contribute to food sovereignty [76].

On the other hand, *Annona purpurea* Moc. & Sessé ex Dunal is a semi-domesticated fruit tree for local consumed in certain regions of Mexico that, as a consequence of the lack of enough information, was classified as an underutilized species. Nevertheless, around 44 alkaloids, 27 essential oils, four flavonoids, and two steroids—which represent potentially bioactive compounds—have been currently identified [77] in this fruit.

Underutilized crops are also candidates for genetic improvement using biotechnological and molecular tools. In this regard, the application of nuclear energy was analyzed to improve native Mexican pseudo cereals—classified as underutilized, but which have high nutritional value—including *Chenopodium berlandieri* ssp. *nuttalliae* (red chia), *Amaranthus hypochondriacus* Aztec race (white chia), and Mixteca race (black chia), as well as *Chenopodium berlanideri* sbp. *nuttalliae* (huauzontle). Genetic improvement was achieved through GR induction of the seeds and a subsequent establishment of the mutant line in the field, where it is expected to achieve improvements, such as reduced dehiscence, larger seed size, large and compact spike, among others [78].

Conservation models must be directly correlated to use; otherwise, there is a risk of losing agrobiodiversity as a consequence of the lack of consumers [79]. Few reports yet include the underutilized species category; however, due to the characteristics of the crops, several researches can be identified in which GR has been used to improve species that fall into this category.

This review gives examples of genetic improvement using GR (Table 2) in crops with high impact on food and plant species which are considered unusual, forgotten, in danger of extinction, of local use, and with a crucial bio prospective potential.

Within this context, a general genetic improvement route with GR and cobalt 60 is proposed, based on the effect they have on different crops (Figure 2). A summary of the methodology with GR in combination with genetic improvement biotechnological techniques is shown. The proposal includes five stages. The first includes aspects that give value to the underutilized species, such as traditional knowledge, morphological characteristics, nutritional contribution, and socioeconomic impact (Figure 2: Knowledge of the species). The use of GR during the second stage (Figure 2: Objectives) depends on the purpose of the research, including: genetic improvement, increased secondary metabolites, phenotypic improvements, tolerance to water stress, salinity, resistance to pests, and increased postharvest life. Once the objectives and scope have been established, we propose using biotechnological techniques, such as in vitro culture and cobalt 60 radiation. In the third stage, the plants should be placed in a greenhouse (Figure 2), to control its environment and protect it from any pollutant, pathogen, or pest. After the mother plant is obtained, the buds or shoots are established in vitro for their multiplication and induction of morphogenesis. Since undifferentiated cells can be worked with, allowing the generation of mutants from the first irradiated generation, indirect morphogenesis guarantees that radiation will have a more significant impact. The fourth stage consists of generating a radiosensitivity curve, to obtain the median lethal dose (LD_50_) or the median growth reduction (GR_50_), and to guarantee the generation of point mutations. Finally, the fifth stage includes plants obtained through regeneration and the evaluation of the morphometric, phytochemical, and physiological differences with the aim of identifying beneficial characteristics.

In this way, the phenomenon of mutagenesis and how a different variety is obtained can be explained. It is essential to highlight that this route is proposed for underutilized plant resources that do not compete in agricultural markets. However, experience indicates some intangible values are essential for society and that conservation actions should be favored [80].

Finally, following the callus irradiation line can reduce the time required to obtain an improved variety. Figure 3 describes the traditional improvement of the variety of *Sechium edule* var. countryside to obtain better quality fruits and longer shelf life. Using the Stratified Mass Selection method, it was possible to obtain these characteristics for the export market in five years [81]. Meanwhile, genetic improvement (biotechnological techniques) and GR were used to obtain the same results in two years (personal communication, unpublished data).

## 8. Conclusions

As a tool for plant breeding, GR has opened several options for food and agriculture. Some reports indicate that the effects that GR generates in specialized organelles and plant cells are aimed at practical applications, promoting new values in cultivars, which also involve a source of nutrients, active pharmaceutical principles, innovative standards for agriculture, or adaptive changes to the environment. A significant number of species which have undergone GR-induced mutations are included among the underutilized varieties—in which genetically improved cultivars have been discovered. The diversity of GR effects results essentially from individual responses and the phenotypic plasticity of each organism. It is important to point out that, although this irradiation technology is based on chance, good results can be obtained if it is directed towards a target and is linked to biotechnological techniques.

## Figures and Tables

**Figure 1 plants-11-01161-f001:**
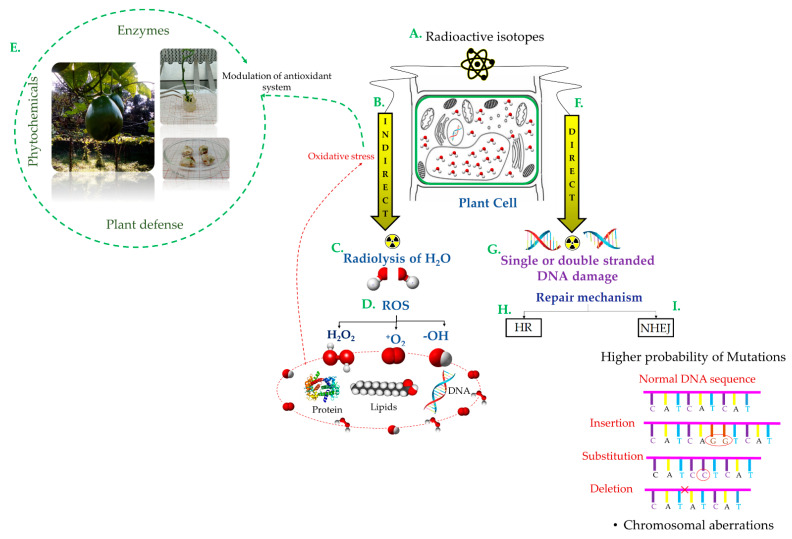
Scheme of Mutation Inductions by gamma radiation for the genetic improvement of plants. (**A**) Gamma radiation (GR) is one of the most widely used mutagenic agents in plants, because it is reported to induce high genetic variability. This type of radiation can be generated by radioisotopes such as carbon-14 (^14^C), cobalt-60 (^60^Co), cesium-137 (^137^Cs), and plutonium-239 (^239^Pu) [1]. Gamma rays (**B**) interact indirectly through (**C**) Radiolysis of water produces (**D**) reactive oxygen species (ROS) (hydrogen peroxide (H_2_O_2_), superoxide anion (O_2_), hydroxyl radical (OH), and singlet oxygen (O=O)) that generate lipid peroxidation and alter the structure of DNA and proteins [2,3,4]. (**E**) By increasing the ROS concentration, oxidative stress triggers the defense of the plant, which is modulated by enzymes such as peroxidase, ascorbate peroxidase, superoxide dismutase, and glutathione reductase [2,6]. Primary gamma radiation lesions delay or inhibit cell division and affect mitotic activity, growth rate or habit, dilation of thylakoid membranes, photosynthesis, modulation of the antioxidant system, and accumulation of phenolic compounds [7]. (**F**) Direct gamma radiation can generate base modifications and (**G**) single or double DNA strand breaks [8]. A twofold mechanism is involved in the natural repair of these errors: (**H**) Homologous Recombination (HR), an error-free repair mechanism; and (**I**) Non-Homologous End-Joining (NHEJ), a mechanism with a greater probability of generating mutations in the repair site, including deletions, insertions, and substitutions, among others [9].

**Figure 2 plants-11-01161-f002:**
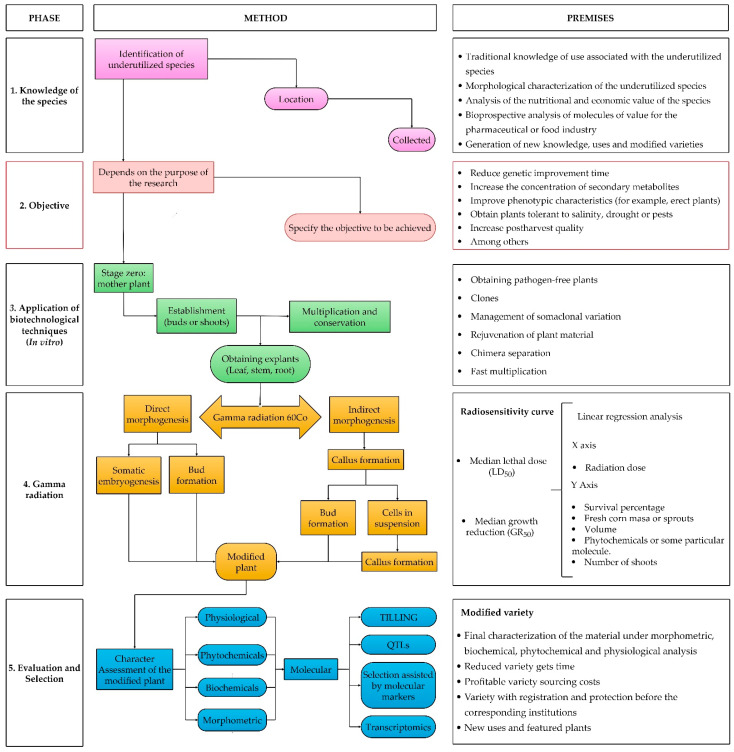
Proposal for the induction of in vitro mutagenesis with cobalt-60 for the genetic improvement of underutilized varieties. 1.—knowledge of the species, 2.—objectives, 3.—application of biotechnological techniques, 4.—Gamma radiation, and 5.—evaluation and selection.

**Figure 3 plants-11-01161-f003:**
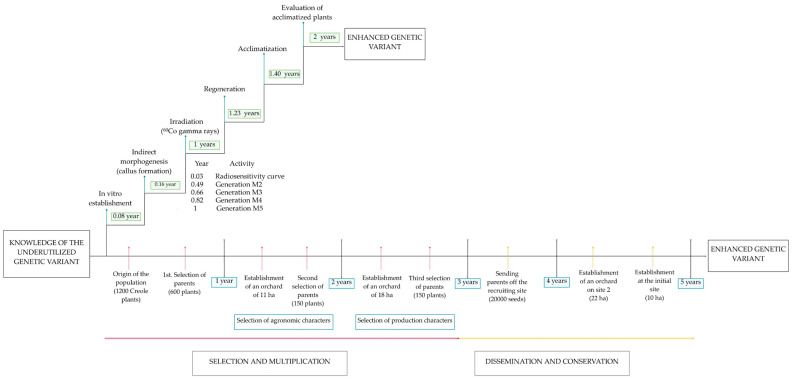
Comparative representation of the time reduction between traditional participatory breeding [81] and GR combined with in vitro techniques for the fruit of *S. edule*.

**Table 1 plants-11-01161-t001:** Priority crops and Network of Conservation Centers in Mexico [75].

Basic and Industrial	Fruit Trees	Vegetables	Impulse	Ornamental
*Agave* spp.	*Persea* spp.	*Capsicum* spp.	*Bixa orellana* L.	Bromeliaceae
*Gossypium barbadense* L.	*Theobroma* spp.	*Solanum lycopersicum* L.	*Suaeda acuminata* (C. A. Mey.) Moq.	Cactaceae
*Phaseolus vulgaris* L.	*Juglans* spp.	*Cucurbita* spp.	*Portulaca oleracea* L.	*Tagetes* spp.
*Helianthus annuus* L.	*Carica* spp.	*Ipomoea* spp.	*Yucca* spp.	*Dahlia* spp.
*Jatropha curcas* L.	*Vitis* spp.	*Sechium* spp.		*Echeveria* spp.
*Zea mays* L.	*Annona* spp.	*Solanum cardiophyllum* Lindl.		*Hymenocallis* spp.
*Vanilla* spp.	*Spondias* spp.	*Physalis* spp.		Euphorbiaceae
*Amaranthus* spp.	*Psidium guajava* L.			Orchidaceae
*Simmondsia chinensis* (Link) C. K. Schneid	*Byrsonima crassifolia* (L.)Kunth			*Beaucarnea recurvata* Lem.
	*Opuntia* spp.			*Tigridia* spp.
	*Philodendron* spp.			
	*Pouteria* spp.			
	*Crataegus* spp.			

**Table 2 plants-11-01161-t002:** Gamma radiation treatment and LD_50_ determination to obtain putative mutant lines of different plant species.

Common Name	Scientific Name	Irradiated Tissue Material	Treatment	LD_50_	Observations	Reference
Watermelon	*Citrullus lanatus* (Thunb.) Matsum. & Nakai var. *Kaolack* and var. *Crimson sweet*	Seeds	100, 200, 300, 400, and 600 Gy	*Kaolack* 225.40 Gy and *Crimson sweet* 221.56 Gy	Radiosensitivity of the two most frequently cultivated varieties in Cameroon and determination of LD_50_.	[16]
Coffee plant	*Coffea arabica* L. var. *typica*	Seeds	0, 50, 100, and 150 Gy	100 Gy	Determination of LD_50_ and morphological changes in plant	[17]
Wilman lovegrass	*Eragrostis superba* Peyr.	Seeds	100, 200, 300, 450, 600, 900, 1400, 2000, and 4000 Gray	2486 Gy	Determination of LD_50_.	[18]
Grasses: llorón, buffel, banderita, and navajita	Lloron (*Eragrostis curvula*)*,* buffel (*Pennisetum ciliare*), banderita (*Bouteloua curtipendula*), and navajita (*Bouteloua gracilis*)	Seeds	100, 200, 300, 450, 600, and 900 Gray	Pasto lloron 628 Gy, buffel 712 Gy, banderita 698 Gy, and navajita 411 Gy	Determination and comparison of LD_50_ in pastures.	[19]
Agave	*Agave tequilana* Weber var. *Azul*	Callus cultures and seedlings	10, 20, 30, 40, and 50 Gy	seedlings 20–25 Gy; Callus 16 Gy	Determination of LD_50_ and comparison between plant material.	[20]
Potato	*Solanum tuberosum* L. var. *Désirée*	Callus cultures	5, 10, 15, 20, and 30 Gy	10 Gy	Determination of mean lethal dose.	[21]
Golden berry/Uchuva	*Physalis peruviana* L.	Axillary buds	50, 100, 200, and 300 Gy		Higher percentage of cells with chromosomal alterations.	[24]
Chrysanthemum	*Chrysanthemum morifolium* (Ramat.) “Donglinruixue”	Seeds	0, 15, 20, 25, 30, and 35 Gy *	35 Gy	The seeds will form genomic and chromosomal abnormalities during anaphase.	[26]
Sugar cane	*Saccharum* spp. Híbrido var. “*SP 70-1284*”	Callus cultures	10, 20, 30, 40, 50, 60, 70, and 80 Gy	30 Gy	Determination of LD_50_.	[27]
Gerbera	*Gerbera jamesonii* H. Bolus	In vitro explant growth, callus cultures and seedlings	10, 20, 30, 40, 50, and 60 Gy	20 gy	Callus fresh weight decrease response.	[30]
Beach purslane	*Sesuvium portulacastrum* L.	Shoots	5 to 40 Gy	20 Gy	Increased concentration of ecdisteroid 20-hydroxyecdysone.	[31]
Orchid	*Dendrobium sonia*	Shoots	15–45 Gy	30 GY	GR decreased shoot length, fresh weight, and leaf area, but its combination with yellow light increased shoot survival and length, fresh weight, and chlorophyll content	[32]
Loquat	*Eriobotrya japonica* L.	Callus cultures and seedlings	(0, 10, 30, and 50 Gy)	10 Gy	Response in growth traits: callus diameter, callus height, number of shoots, number of leaves, and height of seedlings.	[33]
Banana	*Musa paradisiaca* L.	In vitro sprout seedlings	10 Gy, 20 Gy, and 30 Gy	10 and 20 Gy	Seedling morphological properties. Bases of mass propagation.	[34]
Citrus	*Citrus* spp. (several varieties: *‘Alemow’* and *sour orange* as citrus rootstock, lemon cv. *‘Fino 49*’ and ‘*Verna 51*’, tangerine cv. *‘Nova’*, and lime cv. *‘Bearss’*)	Seeds, buds, and nodal segments	Seeds 0, 50, 100, 150, 200, and 250 Gy Buds 0, 25, 50, 75, and 100 Gy Nodal segments 0, 10, 20, 30, 40, and 50 Gy	Seeds (LD_50_ of 127 Gy in *Alemow*, and 156 Gy in *sour orange*). Buds (LD_50_ around 50 Gy for all cultivars) and nodal segments (LD_50_ around 25 Gy for both lemon cultivars).	Difficult-breeding species.	[35]
Wheat	*Triticum aestivum* L.	Seeds	100, 200, 300 and 400 Gy	100 Gy	85% increase in proline concentration and higher chlorophyll a concentration in seedlings.	[36]
Chickpea	*Cicer arietinum* L.	Seeds	50 a 750 Gy (frequency of 50 Gy) with a dose rate of 10.606 Gy min^−1^	150 Gy	Lines resistant to *Ascochyta rabiei*.	[44]
Rice	*Oryza sativa* L. var. *Mentik Susu*	M3 Seeds		200 gGy	Mutants with short plant height, high productivity, higher seed yield, and short harvest age.	[45]
Asiatic spark	*Centella asiática* (L.) Urb.	Axillary buds	0, 10, 20, 30, 40, 50, 60, 70, 80, 90, 100, and 120 Gy	20 and 30 Gy	Higher concentrations of total flavonoids.	[49]
Esparceta, Sainfoin	*Onobrychis viciifolia* Scop. Syn. *Onobrychis sativa* L.	Seeds	30, 60, 90, and 120 Gy	90 Gy	Remarkable increase in the phenolic content of the leaf extract and increase of alkaloid Berberine.	[50]
Barijeh	*Ferula gummosa* Boiss.	Callus cultures	0 to 25 Gy	Of 20 and 25 Gy	Increased phenolic content.	[51]
Jengger Ayam	*Celosia cristata* L.	Seedlings	0, 25, 50, and 75 Gy	25 Gy	The C1U3 2.3.1 mutant presents triterpenic compounds that were not found in the controls.	[52]
Curled-leaved St. John’s-wort	*Hypericum triquetrifolium* Turra	Callus cultures	10, 20, 40, and 50 Gy	10 Gy	Higher content of phytochemicals than in the control samples.	[54]
Fenugreek	*Trigonella foenum-graecum* L.	Seeds	0, 100, 200, 300, and 400 Gy	100 Gy	7% and 9% increases in trigonelline and nicotinic acid.	[55]
Common madder or Indianmadder	*Rubia cordifolia* L.	Callus cultures	2, 4, 6, 8, 10, 12, 14, and 16 Gy	8 Gy	Radiation dose for kinetic study of cell growth and anthraquinone content. They accumulated a maximum level of alizarin and glitter that were 6 and 11 times higher than the non-irradiated callus cultures.	[56]
Barley	*Hordeum vulgare* L.	Seedlings	50–300 Gy	50 Gy	High concentration of proline and antioxidant enzyme activity. Heavy metal stress resistance.	[61]

* Gy-Gray unit of measure (J Kg^−1^): absorption of one joule of radiation energy per kilogram of matter.

## Data Availability

Not applicable.

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
