# Peer review of "Use of Gamma Radiation for the Genetic Improvement of Underutilized Plant Varieties"

_plants, 2022, doi:10.3390/plants11091161_

Round 1

Reviewer 1 Report

The authors have comprehensively reviewed the advancement and importance of the gamma radiation technology in the genetic breeding of various plant species of important agricultural values but generally underutilized.  I believe that the authors have provided sufficient background, included comprehensively the relevant literatures, and concluded appropriately based on available data. The manuscript is written well, in particular, the figures are well structured, informative, and helpful in understanding the topics presented. I have enjoyed reading this manuscript very much. I do not have any technical concerns but a few editorial corrections listed here for the authors to consider if a revision is requested by the editor.

Figure 1, in the caption, please change (A.), (B.), (C.), etc., to (A), (B), (C), etc…

Line 51, [2,3,4] or [2-4] but not [3,4,2]

Line 54, change “[6,2]” to [2,6]

Line 97, typica needs to be italicized.

Lines 219 and after, to the end of this manuscript, many short paragraphs could be actually combined into a few longer paragraphs…

Line 262, I believe that the reference is cited using an incorrect format, and actually many references are cited using this incorrect format, please consider.

Table 1, please be careful of the use of “spp.” and make sure it is not “sp.” for just one species. Also, the space between rolls is strange, are you trying to align each roll or not really?

Table 2, I see many incorrect formats, i.e., the italicized author names of the scientific names but they should not have been, and maybe some var. names (e.g., typica) should have been italicized but they are not…please proofread this part in this table carefully.

Figure 2, the box labeled with “2. Objective” is extremely large, and looks proportionally awkward in comparison to the rest of this figure. Please make some adjustment.

Section 8. Conclusions, again, I believe that these three short paragraphs could be combined into one single paragraph.

Lines 553 and after, apparently, the authors forgot the information for these sections…

Reviewer 2 Report

This review paper describes that the use of gamma irradiation is a good option to improve underutilized plants in a relatively short time. I think the manuscript is well structured and clearly written. The following points needs to be considered before accepting the manuscript.

Line 59: Why “(MR)” is the abbreviation for an error-free repair mechanism?

Line 59: I am not familiar with the term “NO Homologous Recombination (NHR)”. I believe this pathway is commonly known as “non-homologous end joining (NHEJ)”. Please confirm the sentence in Line 123, too.

Line 96: Spell out “LD50” because it first appears in the text. I think this is NOT an abbreviation for “a radiosensitivity curve (Line 513)”.

Line 122: double strand break (DSB) repair?

Line 202: A comma is missing; “radiation, drought or saline stress”

Line 318: Magdy et al. [53] recorded >>> .

Line 395: “proton beam treatment”?

Definition of the endpoints: The endpoints used in the manuscript are unclear and confusing. The authors often use “LD50” in the main text, while “IC50”, “DL50”, “DCL50” and “GR50” are used in Table 2 and Figure 2. Please specify all of them somewhere in the main text, and also describe how the different endpoints are used in this research field. I believe this is an important information for the readers.
